An examination of species limits in the Aulacorhynchus “prasinus” toucanet complex (Aves: Ramphastidae)

Winker Kevin kevin.winker@alaska.edu
University of Alaska Museum, University of Alaska Fairbanks
Pie Marcio
Electronic publication date: 2016 Aug 30
Publication date: 2016
Volume: 4
Electronic Location ID: e2381
Received 2016 May 13; Accepted 2016 Jul 30
Copyright: ©2016 Winker
Copyright year: 2016
Copyright holder: Winker
License: This is an open access article distributed under the terms of the Creative Commons Attribution License, which permits unrestricted use, distribution, reproduction and adaptation in any medium and for any purpose provided that it is properly attributed. For attribution, the original author(s), title, publication source (PeerJ) and either DOI or URL of the article must be cited.
License URL: https://creativecommons.org/licenses/by/4.0/

Keywords: Middle America, Morphometrics, Taxonomy, South America, Neotropics

Funding: Chapman Memorial Fund Visit to AMNH was supported in part by a grant from the Chapman Memorial Fund. The funders had no role in study design, data collection and analysis, decision to publish, or preparation of the manuscript.

==============================
The number of species recognized in Aulacorhynchus toucanets has varied tremendously over the past century. Revisors seem to disagree on whether head and bill coloration are useful indicators of species limits, especially in the A. “prasinus” complex. Using morphometrics, I tested the hypothesis that the major color-based subspecific groups of A. “prasinus” sensu lato are simply “cookie-cutter” (i.e., morphologically nearly identical) toucanets with different head and bill colorations. Univariate and multivariate analyses show that they are not simply morphological replicates of different colors: a complex array of morphometric similarities and dissimilarities occur between the major subspecific groups, and these variations differ between the sexes. Latitude and longitude had a small but significant association with female (but not male) PC1 and PC2. Hybridization and intergradation were also considered using plumage and bill characters as a surrogate to infer gene flow. Hybridization as indicated by phenotype appears to be substantial between A. “p.” cyanolaemus and A. “p.” atrogularis and nonexistent between other major groups, although from genetic evidence it is likely rare between A. “p.” albivitta and A. “p.” cyanolaemus. The congruence and complexities of the morphological and color changes occurring among these groups suggest that ecological adaptation (through natural selection) and social selection have co-occurred among these groups and that species limits are involved. Further, hybridization is not evident at key places, despite in many cases (hypothetical) opportunity for gene flow. Consequently, I recommend that this complex be recognized as comprising five biological species: A. wagleri, prasinus, caeruleogularis, albivitta, and atrogularis. Four of these also have valid subspecies within them, and additional work may eventually support elevation of some of these subspecies to full species. Species limits in South America especially need more study.

Introduction

In spite of ongoing advances in the description and recognition of biodiversity, few genera can offer such an incongruous history as Aulacorhynchus Gould (Aves: Piciformes: Ramphastidae). The Aulacorhynchus toucanets inhabit montane forests from Mexico to Guyana and Bolivia, and there are many allopatric taxa. Although generic limits have been generally consistent during the past century, the number of species within the genus has been a matter of considerable disagreement.

Taxonomic history

In the second edition of his monograph on the Ramphastidae, Gould (1854) recognized 10 species in the genus Aulacorhamphus (now Aulacorhynchus by priority), but several taxa remained undescribed at that time. Between them, Salvin & Godman (1896) and Brabourne & Chubb (1912) recognized 15 species in the genus. Ridgway (1914) followed this treatment. Cory (1919) considered one of these species (A. erythrognathus) to be only a subspecies, and he treated the genus as having 14 species. Although a new species was described in 1933 (A. huallagae, Carriker Jr, 1933), Peters (1948) reduced many of the formerly recognized species to subspecific status and presented the species-level diversity of the genus as just seven taxa. Haffer (1974) followed Peters (1948), except that he reduced one species (A. calorhynchus) to subspecies status (after Schwartz, 1972), leaving just six. Sibley & Monroe Jr (1990) followed this treatment, but they presented a further reduction of the apparent diversity by mentioning only two subspecific groups below the species level. Although more comprehensive subspecific inclusion and discussion were given by Short & Horne (2001) and Short & Horne (2002), six full species were retained. The treatment of this genus since 1891 is summarized in Table 1.

Table 1 Treatments of species-level diversity in the genus Aulacorhynchus.

Taxa historically recognized only as subspecies are not included (see text for these taxa in “prasinus”). An X means the taxon was treated as a species, a dash indicates not available to be treated yet, and a blank indicates that the taxon was not considered.

	Sclater(1891)	S & G (1896)a B & C (1912)b	Cory (1919)	Peters (1948)	Sibley & Monroe (1990)	Short & Home (2001)	Nav. et al. (2001)g P-O et al. (2008)g B. et al. (2011)g	Dickinson & Remsen (2013)i	Del Hoyo & Collar (2014)	this article	
A. sulcatus	×	×	×	×	×	×	×	×	×		
A. erythrognathus	×	×	ssp. of sulcatus	ssp. of sulcatus		ssp. of sulcatus	ssp. of sulcatus	ssp. of sulcatus	ssp. of sulcatus		
A. calorhynchus	×	×	×	×	ssp. of sulcatus	ssp. of sulcatus	ssp. of sulcatus	ssp. of sulcatus	×		
A. derbianus	×	×	×	×	×	×	×	×	×		
A. whitelianus	×	×	×	ssp. of derbianus		ssp. of derbianus	×	×	×		
A. haematopygus	×	×	×	×	×	×	×	×	×		
A. caeruleicinctis	×	×	×	×	×	×	×	×	×		
A. huallagae	−c	−c	−c	×	×	×	×	×	×		
A. prasinus	×	×	×	×	×	×	×	×	×	×	
A. wagleri	×	×	×	ssp. of prasinus		ssp. of prasinus	×	ssp. of prasinus	×	×	
A. caeruleogularis	×	×	×	ssp. of prasinus	ssp. of prasinus	ssp. of prasinus	×	ssp. of prasinus	×	×	
A. cognatus	−d	−d	ssp. of caeruleogularis	ssp. of prasinus		ssp. of prasinus	×	ssp. of prasinus	ssp. of caeruleogularis	ssp. of caeruleogularis	
A. albivitta	×	×	×	ssp. of prasinus		ssp. of prasinus	×	ssp. of prasinus	×	×	
A. griseigularis	−e	−e	−e	ssp. of prasinus		ssp. of prasinus	×	ssp. of prasinus	ssp. of albivitta	ssp. of albivitta	
A. lautus	−f	×	×	ssp. of prasinus		ssp. of prasinus	×h	ssp. of prasinus	ssp. of albivitta	ssp. of albivitta	
A. cyanolaemus	×	×	×	ssp. of prasinus		ssp. of prasinus	ssp. of atrogularis	ssp. of prasinus	×	ssp. of atrogularis	
A. dimidiatus	×	×	×	ssp. of prasinus		ssp. of prasinus	ssp. of atrogularis	ssp. of prasinus	ssp. of atrogularis	ssp. of atrogularis	
A. atrogularis	×	×	×	ssp. of prasinus		ssp. of prasinus	×	ssp. of prasinus	×	×	
Notes.

a Salvin & Godman (1896) treated only Middle American Aulacorhynchus, which at the time were considered Aulacorhamphus.

b Brabourne & Chubb (1912) treated South American members of the genus (then considered Aulacorhamphus).

c huallagae was described by Carriker Jr (1933).

d cognatus was described as a subspecies by Nelson (1912).

e griseigularis was described as a subspecies by Chapman (1915).

f lautus was described by Ban (1898).

g Navarro et al. (2001), Puebla-Olivares et al. (2008) and Bonaccorso et al. (2011) together included most Middle American and South American Aulacorhynchus taxa.

h though not included in either study.

i Treatment matches the South American Classification Committee (Remsen Jr et al., 2016).

The massive lumping of Peters (1948) proceeded with neither the presentation of data nor with discussion. Careful study of one taxon, A. calorhynchus, by Schwartz (1972) supported the single species-level change that was made between Peters (1948) and Haffer (1974), and Haffer’s (1974) important work has been cited to support maintaining a broad A. prasinus (sensu lato; AOU, 1998). Haffer (1974), Short & Horne (2001) and Short & Horne (2002) used evidence of hybridization and intergradation to support their conclusions that the whole A. “prasinus” complex represented one biological species with many subspecies (14 and 13 subspecies, respectively); at the species level across the genus Short & Horne (2001), Short & Horne (2002) treatment reflected that of Dickinson & Remsen Jr (2013) in Table 1 except that they considered A. whitelianus a subspecies of A. derbianus, as Peters (1948) did. Presumably similar reasoning was behind Peters (1948). But subspecies, even those distinctive enough to have been considered full species for a century, can get lost in the shuffle. For example, oversimplification of subspecific variation led Sibley & Monroe Jr (1990) and the American Ornithologists’ Union (AOU, 1983; AOU, 1998) to completely omit mention of the very distinct form A. “prasinus” wagleri from southwestern Mexico. The AOU (1983; 1998) considered Middle American diversity in the genus as being just two subspecific groups of a single species, A. prasinus.

Renewed interest in this complex (Navarro et al., 2001; Puebla-Olivares et al., 2008; Bonaccorso et al., 2011; Del Hoyo & Collar, 2014) is beginning to rectify the absence of data, but the ensuing taxonomic changes recommended have either been based on a different species concept (Bonaccorso et al., 2011) or have inadequately considered the hybridization and intergradation (e.g., Navarro et al., 2001; Puebla-Olivares et al., 2008; Del Hoyo & Collar, 2014) that have been integral to supporting the “post-Peters” taxonomy. These latter works have recommended elevation of numerous A. “prasinus” taxa to species status (Table 1), but they did not address the reasons for lumping in the first place: evidence of hybridization. There has also been heavy reliance on a single molecular marker (mtDNA) for species delimitation in A. “prasinus” (Puebla-Olivares et al., 2008; Bonaccorso et al., 2011). This is problematic because mtDNA can be misleading about species limits and relationships between populations due to gene-tree/species-tree mismatches and because genetic distance is not a reliable indicator of species limits (Avise & Wollenberg, 1997; Irwin, 2002; Funk & Omland, 2003; Degnan & Rosenberg, 2006; Cheviron & Brumfield, 2009; Galtier et al., 2009; Ribeiro, Lloyd & Bowie, 2011; Toews & Brelsford, 2012; Pavlova et al., 2013; Peters et al., 2014; Dolman & Joseph, 2015; Morales et al., 2015). Thus, species limits in the group remain uncertain (Table 1). Most disagreement has been in the A. “prasinus” complex, and it is on this group that I focus.

The A. “prasinus”complex.—As currently treated (e.g., Table 1, Dickinson & Remsen Jr, 2013; Del Hoyo & Collar, 2014), A. “prasinus” either is a widely distributed and highly variable biological species or it comprises multiple species (Fig. 1). So far as is known, Aulacorhynchus toucanets are nonmigratory (AOU, 1998), but as O’Neill & Gardner (1974) and Navarro et al. (2001) noted, members of the genus can wander widely during the nonbreeding season. Short & Horne (2001) considered Central American forms from Mexico to Panama to be partially migratory, with downslope movements to lowlands also occurring during the nonbreeding season (but less commonly) throughout the A. “prasinus” range. The sexes are alike by plumage (sexually monochromatic), but sexual size dimorphism is apparent in all taxa examined.

Figure 1 The six major, color-based taxonomic groups of the Aulacorhynchus “prasinus” species complex, from top to bottom.

(A) wagleri; (B) prasinus (nominate prasinus and warneri, the full-bodied bird, are portrayed): (C) caeruleogularis; (D) albivitta (griseigularis and nominate albivitta are portrayed); (E) cyanolaemus (yellow-tipped bill); and (F) atrogularis. Artwork used with permission from Kristin Williams under CC-BY-NC.

In past work there has been too little discussion of the fact that different levels of differentiation occur among the subspecies of A. prasinus, sensu lato. All of the named forms do not represent equally differentiated populations; there are major subspecific groups of one or more described subspecies. Objectively determining what these groups are can be done by following taxonomic history, coupled as it is with a color-based clustering. Cory (1919) recognized eight species that were later lumped by Peters (1948) into A. prasinus (Table 1). It seems that four of these (A. wagleri, lautus, cyanolaemus, and dimidiatus) were not available for Cory (1919) to examine, however (he noted which taxa were in the Field Museum of Natural History at that time). Two of these taxa, A. lautus and A. dimidiatus, may have been included as full species through inertia. This was not uncommon: Ridgway (1914) considered that the genus had 15 species, but he was uncertain because he had only been able to examine seven of them. The subsequent rediscovery and examination of A. dimidiatus (O’Neill & Gardner, 1974) showed intergradation with A. “prasinus” atrogularis, and the Santa Marta isolate A. “prasinus” lautus is, by plumage, clearly a relatively minor derivative of the A. “p.” albivitta group. These considerations reduce the number of major, color-based subspecific groups in the A. “prasinus” complex to six (A. “p.” wagleri, prasinus, caeruleogularis, albivitta, cyanolaemus, and atrogularis).

The rather pronounced differences among these major groups are illustrated in Fig. 1, together with examples of some of the less-pronounced variation (though still between named subspecies) occurring within two of these groups. Del Hoyo & Collar (2014) presented an analysis of colors that reaffirms this approach, coming back to the same six color-based subspecific groups and treating them as full biological species. Diagnostic characters of these six groups are given in Table 2. It is these six groups that form the basis for my comparisons. They are the color-based groupings that have been recognized by students of the birds themselves.

Table 2 Color and pattern-based diagnostic characteristics of the six major subspecific groups of Aulacorhynchus “prasinus” (based on specimens and Del Hoyo & Collar, 2014).

Middle American forms are at left; South American forms are at right.

A.“p.” wagleri (monotypic): greatly enhanced orange, paedomorphic bill encrustations at the very base of upper mandible; orange band at base of lower mandible; broad black base of upper mandible; yellowish-white forehead grading to olive crown; pale bluish-green underparts.	A. “p.” albivitta (incl. subspp. A. “p.” lautus, griseigularis, phaeolaemus, and albivitta): yellowish skin surrounds more than half the eye; variable chin and throat (white, grayish, pale gray-blue); chestnut at base of lower mandible in most.	
A. p. prasinus (incl. subspp. A. p. prasinus, warneri, virescens, andvolcanius): upper mandible broadly yellow to base; black patches at nares; bright yellow stripe at base of mandible; white cheeks below eye.	A. “p.” cyanolaemus (monotypic): upper mandible mostly black; pinkish skin around part of eye; deep blue-gray chin and upper throat with little to none on cheek.	
A. “p.” caeruleogularis (incl. subspp. A. “p.” caeruleogularis and cognatus): deep rich blue chin, throat, and cheek; rich olive crown and nape in absence of pale throat.	A. “p.” atrogularis (incl. subspp. A. “p.” atrogularis and dimidiatus): strongly bi-colored, orange-yellow vs. white basal bill stripe going from upper to lower mandibles; black chin, throat, and cheek; dark skin around eye (only one of group in South America).	

The troubled taxonomic history of Aulacorhynchus (especially A. prasinus, sensu lato) reflects disagreement among revisors over the ability of plumage and bill colors and patterns to represent species limits. Members of the A. “prasinus” group in particular are effectively ecologically similar geographic replacements (Mallet, 2007), and where named taxa have been found to come together hybridization has been described (e.g., Haffer, 1974; Short & Horne, 2001). Given evidence of hybridization, it is unclear how the birds themselves perceive these differences. Is this group (A. “prasinus”sensu lato) really one in which head and bill colors and patterns are very plastic, resulting in species that include a high degree of color variation? Or are full species being overlooked? If bill and head color characteristics are plastic and not indicative of species limits, as many treatments since 1948 suggest, then strikingly color-based taxa, most of Cory’s (1919) species and the six of Del Hoyo & Collar (2014), would not likely show a great deal of morphometric distinctiveness. If, on the other hand, substantial, concordant morphological changes are also occurring, then perhaps the concept of ecologically similar geographic replacements, which seems broadly applicable to A. prasinus, sensu lato, is masking important group-specific evolutionary adaptations that go beyond the existing color changes, the latter of which likely reflect social selection. Such morphological differences might suggest adaptive changes among groups that would make immigrants and hybrids less fit (Price, 2008). My question then is simple: Are these color-based subspecific groups just “cookie-cutter” (i.e., morphologically nearly identical) toucanets bearing different throat and bill colorations, or are there also significant morphological changes occurring among them? If substantial morphometric changes occur concurrently with dramatic color changes, then species limits should probably be reconsidered, as several have suggested (Navarro et al., 2001; Puebla-Olivares et al., 2008; Bonaccorso et al., 2011; Del Hoyo & Collar, 2014).

Morphometrics alone are unlikely to be important components of species limits in forms like these where colors are obviously important, and this is not meant as a study of how morphology varies within the group independently of color-based clustering. Navarro et al. (2001) did a morphological analysis based on 17 allopatric groupings, and this is not meant to repeat those analyses. Geography alone can affect morphology (e.g., Bergmann’s rule; Mayr, 1963), and my analyses account for this. In this study I will (1) test for univariate differences between pairwise groups that are geographically closest to each other; (2) compare these groups in multivariate, principal component space (because univariate measures can be correlated with each other); and (3) visually examine specimens for evidence of hybridization because such evidence has been historically important in the taxonomy of the group.

Methods and Results

I used morphometrics to examine how body characteristics vary among the major, color-based subspecific groups in the Aulacorhynchus “prasinus” complex (Fig. 1, Table 2). Museum specimens (institutions listed in Acknowledgments) were visually examined and measurements of wing chord, tail, tarsometatarsus, bill, bill height, and bill width (all three bill measures from anterior edge of nares) were made to the nearest 0.1 mm using vernier calipers following Baldwin, Oberholser & Worley (1931). Wing tip (length of longest primary to first secondary) was also measured to the nearest 0.1 mm using vernier calipers. Although Navarro et al. (2001) raised the possibility that bill growth in this group might be indeterminate, with my larger sample size of 98 male A. prasinus sensu stricto (see Appendix) I found that bill length had a normal distribution, so it is retained in my analyses. Morphometric geographic variation within these six major subspecific groups was not examined, because that is not related to the hypothesis being tested, i.e., are morphological changes occurring concurrently with color-based changes (see also Navarro et al., 2001). However, the effects of geography upon the data are examined after the main questions posed are addressed. Some of these major subspecific “groups” have named subspecies within them. Aulacorhynchus prasinus has four (prasinus, warneri, virescens, volcanius), A. “p.” caeruleogularis two (caeruleogularis, cognatus), A. “p.” albivitta four (lautus, griseigularis, phaeolaemus, albivitta), and A. “p.” atrogularis two (atrogularis, dimidiatus); the other two major subspecific groups (wagleri, and cyanolaemus) have no named subspecies within them (Appendix). Color differences formed the basis for the majority of characters used to describe all of these named subspecies, with size being mentioned (in addition to color) in just three of 14 cases. Within-group variation is accounted for in the standard statistical manner (e.g., Table 3). Immature individuals were not measured.

Table 3 Mensural characteristics of both sexes among six major groups of the Aulacorhynchus “prasinus” complex.

Units are mm except for mass (g) and sample size (N).

		prasinus	wagleri	caeruleogularis	albivitta	cyanolaemus	atrogularis	
		M	F	M	F	M	F	M	F	M	F	M	F	
Mass (g)	mean	186.6	171.2	169.7	172.0	152.8	158.5	–	–	176.7	158.8	158.1	112.5	
s.d.	20.98	24.78	14.10	15.94	24.67	5.72	–	–	23.57	28.53	25.23	2.50	
min	153.6	135	145	127	118	154	–	–	160	130	124	110	
max	239.2	229.6	200	189.3	184	166.6	–	–	210	210	188	115	
N	13	11	13	14	8	3	–	–	3	5	8	2	
Wing chord (WCH)	mean	124.1	119.6	125.9	123.3	120.0	116.5	128.1	125.5	127.1	125.0	122.4	118.0	
s.d.	4.58	4.95	3.80	3.64	4.90	4.38	4.26	3.24	5.78	3.62	6.07	7.00	
min	113.1	106.4	117.4	117.4	103.5	104.0	118.6	117.7	114.0	119.9	108.1	105.5	
max	135.0	130.7	136.7	132.3	133.1	125.6	139.5	132.4	137.8	131.2	133.6	129.6	
N	98	74	28	26	105	50	86	57	9	10	17	11	
Tail (TL)	mean	109.7	105.0	113.1	111.8	98.4	94.6	109.8	106.4	106.8	109.1	112.2	106.3	
s.d.	6.01	6.48	5.15	4.42	5.04	4.60	6.90	5.90	2.14	6.92	4.89	7.62	
min	95.9	90.8	101.8	103.8	84.3	85.1	92.1	91.5	102.1	96.4	102.9	94.1	
max	124.4	119.0	122.7	121.9	116.1	106.9	127.0	120.5	109.2	118.6	119.8	118.0	
N	94	71	28	25	99	48	84	57	8	10	16	11	
Tarso-metatarsus (TS)	mean	32.4	31.5	32.5	31.7	32.3	31.3	32.9	32.0	32.8	31.5	31.4	29.8	
s.d.	1.45	1.44	0.99	0.99	1.33	1.24	1.29	1.18	1.26	1.46	2.19	1.95	
min	28.1	27.9	29.0	29.7	29.2	29.0	29.4	28.7	31.1	28.8	26.8	26.7	
max	35.8	35.1	34.6	33.4	35.7	34.8	35.5	35.2	35.0	34.2	35.6	33.6	
N	98	75	28	26	105	50	86	57	9	10	17	11	
Bill (BL)	mean	71.2	57.9	71.4	61.2	58.8	48.6	64.5	52.9	65.4	54.8	68.0	54.9	
s.d.	5.75	4.62	4.70	3.77	3.82	3.97	4.96	4.29	5.18	2.15	5.57	6.45	
min	58.4	49.1	64.0	55.2	45.2	42.1	54.3	43.8	55.1	50.8	59.9	46.7	
max	84.7	71.9	84.6	69.7	69.2	68.1	73.3	68.0	75.1	59.4	79.8	69.2	
N	98	75	28	26	105	50	86	57	9	10	17	11	
Bill height (BLH)	mean	24.2	22.7	23.3	22.6	22.3	20.9	23.3	21.7	23.1	22.4	24.7	22.5	
s.d.	1.13	1.18	0.99	0.89	1.00	0.99	1.05	1.04	1.95	0.65	2.28	1.78	
min	20.8	20.7	20.9	21.3	19.6	19.1	20.2	19.3	18.9	21.2	22.9	20.0	
max	26.7	25.1	25.7	25.0	25.2	23.6	25.7	24.2	25.0	23.5	33.1	26.2	
N	97	75	28	26	104	50	86	57	9	10	17	11	
Bill width (BLW)	mean	21.1	20.1	21.3	21.1	21.0	20.2	21.5	20.4	20.4	19.5	21.3	19.7	
s.d.	1.01	0.99	0.72	0.94	1.06	1.07	1.06	1.03	1.13	0.77	1.06	1.50	
min	18.7	17.7	20.0	19.0	17.8	18.2	19.0	17.7	18.3	18.5	19.2	17.4	
max	23.4	23.5	22.7	23.4	23.5	24.0	24.4	22.9	22.4	21.1	22.8	22.8	
N	98	75	28	26	105	50	86	57	9	10	17	11	
Wing tip (WGTP)	mean	16.9	16.7	18.2	17.9	16.8	16.4	16.6	15.9	16.7	16.6	17.0	16.5	
s.d.	2.56	2.78	2.15	1.94	2.06	2.37	2.24	1.87	1.32	1.77	3.02	2.13	
min	11.4	9.9	13.2	13.3	10.8	9.7	11.5	12.1	14.3	14.3	11.2	11.4	
max	24.0	23.0	22.8	21.3	22.8	21.9	22.5	20.0	18.1	19.4	22.6	19.2	
N	95	69	28	26	104	49	86	57	9	10	17	11	

I examined and measured 578 specimens of the six major subspecific groups of A. “prasinus.” The distributions of these groups (Fig. 2) were found to be allopatric or parapatric, as others have depicted (e.g., Haffer, 1974; Short & Horne, 2001; Ridgely & Greenfield, 2001; Restall, Rodner & Lentino, 2006). Morphometric data exhibited male-biased sexual size dimorphism (Table 3), so all analyses were performed separately for each sex.

Figure 2 Distributions of the specimens of Aulacorhynchus “prasinus” examined in this study with the focal six major subspecific groups labeled.

Neither all specimens in existence nor observation records are included, so ranges are not complete. Red stars indicate evidence of hybridization; the top-most one, in Ecuador, is from the study of Puebla-Olivares et al. (2008).

Univariate mensural characteristics (Table 3) were visually examined to determine whether it was warranted to apply statistical testing for differences. This was done to reduce the overall number of tests made, which enhances the power of individual tests when applying multiple-test corrections. No statistical tests were done on mass (due to small sample sizes), and tests were applied in a pairwise manner between groups most proximate to each other (except for A. “prasinus” albivitta-atrogularis). The Bonferroni-style of multiple-test correction is highly conservative, so I did not use it; in controlling for table-wide Type I error (rejecting the null hypothesis when it is true), it raises the likelihood of Type II error (incorrectly accepting the null hypothesis) at the level of the single test (Sokal & Rohlf, 1995; Whitlock & Schluter, 2009). While false discoveries will accrue with multiple testing, determining whether there are differences at the individual test level is very important for a study of this type. I used an approach more commensurate with this need, one that controls for the expected proportion of falsely rejected null hypotheses, the “false discovery rate” (Benjamini & Hochberg, 1995). I report both aspects (uncorrected and corrected), because future investigators of subsets of these taxa should focus on characteristics that differ between them and not be distracted by the additional statistical gyrations that I needed to perform to reduce table-wide error when making so many tests (60 tests for Table 4 and 24 for Table 5).

Table 4 Patterns of significance from results of t-tests of mensural characters between geographic pairs of major subspecific groups of Aulacorhynchus “prasinus.”

Positive (+) values indicate that the first named group averages larger, while negatives (−) indicate that the second is the larger. Character abbreviations follow Table 3.

Pairs compared	Sex	WCH	TL	TS	BL	BLH	BLW	WGTP	N	
prasinus-wagleri	M		**(−)			*** (+)		* (−)	98, 28	
F	** (−)	*** (−)		*** (−)		** (−)		75, 26	
prasinus-caeruleogularis	M	*** (+)	*** (+)		*** (+)	*** (+)			98, 105	
F	*** (+)	*** (+)		*** (+)	*** (+)			75, 50	
caeruleogularis-albivitta	M	*** (−)	*** (−)	** (−)	*** (−)	*** (−)	** (−)		105, 86	
F	*** (−)	*** (−)	** (−)	*** (−)	*** (−)			50, 57	
albivitta-cyanolaemus	M		* (+)				* (+)		86, 9	
F					* (−)a	* (+)		57, 10	
cyanolaemus-atrogularis	M		** (−)						9, 17	
F	* (+)		* (+)a					10, 11	
albivitta-atrogularis	M		** (+)		* (+)	* (−)	* (−)		86, 17	
F	** (+)		** (+)					57, 11	
Notes.

* P < 0.05.

** P < 0.01.

*** P < 0.001.

a Not significant after table-wide correction for false discovery rates (see text).

Table 5 Patterns of significance from results of t-tests comparing individual principal component (PC) scores between geographic pairs of major subspecific groups of Aulacorhynchus “prasinus.”

PC scores are from the first two principal components. Individuals with missing values were excluded. Underlined asterisks indicate significance after false discovery rate correction for multiple tests.

Pairs compared	Sex	PC1	PC2	N	
prasinus-wagleri	M	*		90, 28	
F	***	*	65, 25	
prasinus-caeruleogularis	M	**	***	90, 98	
F	***		65, 48	
caeruleogularis-albivitta	M		***	98, 84	
F	***	**	48, 57	
albivitta-cyanolaemus	M			84, 8	
F			57, 10	
cyanolaemus-atrogularis	M		*	8, 16	
F			10, 11	
albivitta-atrogularis	M		*	84, 16	
F			57, 11	
Notes.

* P < 0.05.

** P < 0.01.

*** P < 0.001.

Significant univariate mensural differences were found among the six groups in both sexes (Tables 3 and 4). The number of significant differences was highest when A. “prasinus” caeruleogularis was compared with A. “p.” prasinus to the north and west (8 differences) and A. “p.” albivitta to the south and east (11 differences; Table 4). After multiple-test correction (which only affected A. “p.” cyanolaemus comparisons in Table 4), the fewest differences occurred when A. “p.” cyanoleamus was compared with A. “p.” albivitta to the north (3 differences) and A. “p.” atrogularis to the south (2 differences), although small sample sizes were likely to be at least partially responsible for this. I included a pairwise comparison between A. “p.” albivitta and A. “p.” atrogularis because of the uncertain taxonomic status of (major/minor group or species/subspecies; Table 1), and small sample sizes available for, A. “p.” cyanolaemus. Intermediate levels of univariate differences occurred between A. “p.” prasinus and A. “p.” wagleri (7 differences) and A. “p.” albivitta and A. “p.” atrogularis (6 differences). Wingtip, bill width, and tarsometatarsus showed the fewest significant differences between groups, whereas wing chord and tail lengths showed the most (Table 4). A pronounced large-small-large pattern was revealed among A. “p.” prasinus-caeruleogularis-albivitta (Tables 3 and 4). The characteristics exhibiting significant differences between taxon pairs varied among pairs and, in most cases, between sexes (Table 4). In other words, significant mensural differences were decidedly inconsistent between groups.

Morphometric relationships between groups (within sexes) were further explored using principal components analyses (PCA). Two analyses were performed. For each sex, all individuals of all groups were run through a single analysis, and the first two principal components (PC1 and PC2) were extracted from the variance–covariance matrix of the log-transformed data. PC1 and PC2 explained 45.0% and 32.4% of the variance among males and 48.0% and 31.1% among females, respectively. For each of the two sex-specific analyses, principal components scores were generated for each individual on PC1 and PC2, and these individual scores were then compared between the major subspecific groups using t-tests. These tests were done to determine whether, on a multivariate basis, morphometric differences between taxon pairs were as heterogeneous as suggested by univariate analyses (Tables 3 and 4). Results suggest that they were; again, differences between groups varied in an unpredicatable manner between the sexes and between the two independent multivariate dimensions (PC1 and PC2; Table 5). Of the multivariate pairwise comparisons, only A. “prasinus” albivitta-cyanolaemus showed no significant differences (Table 5), although several univariate differences were found (Table 4). These results may reflect the small sample size in A. “p.” cyanolaemus. After multiple-test correction, contrasts within the South American forms (the last three rows in Table 5) yielded no significant differences at the table-wide level. Again, while smaller sample sizes likely affected these last results, a “cookie-cutter” effect is not apparent among the major subspecific groups of the A.“prasinus” complex, either in univariate or in multivariate morphometric space (Tables 4 and 5).

The major, color-based subspecific groups of A.“prasinus” do show considerable morphometric differences between them, but are these differences just an expected result from changes of size with latitude under Bergmann’s rule (Mayr, 1963) or otherwise geographically driven? Variation in the two sex-specific principal components was examined in two multiple regressions for each sex (PC1 & PC2) against the variables latitude and longitude. Neither regression was significant in males, but both were in females (F > 4.0, P < 0.02). However, only a small proportion of female variation was explained by latitude and longitude, 4% for PC1 (R2 = 0.04) and 6% for PC2 (R2 = 0.06). Thus, geography has a small but significant influence in 5 of the 11 differences denoted in Table 5, perhaps contributing to the higher significance levels found in females there. Finally, in considering the individual effects of these two geographic variables, stepwise multiple regression showed that only longitude was significantly associated with PC1 in females (F = 7.56, P = 0.007, R2 = 0.039); neither variable by itself was significantly associated with PC2. Thus, there is no evidence that Bergmann’s rule is affecting this complex as a whole.

Hybridization

Because members of the genus are known to wander rather widely during the nonbreeding season, the opportunity for gene flow does exist between these largely allopatrically breeding groups. Among the six major subgroups I examined there are theoretically five pairwise instances of possible gene flow between any two of the groups, particularly across some of the narrower zones of separation, A. “prasinus” prasinus-wagleri (Oaxaca, Mexico), A. “p.” caeruleogularis-albivitta (in W Colombia), A. “p.” albivitta-cyanolaemus (Ecuador), and A. “p.” cyanolaemus-atrogularis (Peru); the fifth, prasinus-caeruleogularis (in Nicaragua), is a larger distance, on the order of about 240 km. Note that closest approach distances are not accurate in Fig. 2, which is based on the specimens I examined; ranges given by other sources (e.g., Haffer, 1974; Hilty & Brown, 1986; Binford, 1989; Howell & Webb, 1995; Short & Horne, 2001; Ridgely & Greenfield, 2001; Restall, Rodner & Lentino, 2006; Schulenberg et al., 2007) include more records (including sight records) and are more accurate.

Figure 3 An example of a hybrid A. “p.” atrogularis× A. “p.” cyanolaemus.

(A) a pure A. “p.” cyanolaemus (LSU 87627); (B) a hybrid (LSU 92029); and (C) a pure A. “p.” atrogularis (LSU 73933).

Specimens were examined carefully for phenotypic evidence (i.e., intermediate phenotypes in plumage and bill coloration) of hybridization between these major groups, but it was found to occur in just one of these pairwise comparisons: between A. “prasinus” cyanolaemus and A. “p.” atrogularis in Peru. Four specimens representing possible F1 hybrids (due to intermediacy of characters) were found; one from La Libertad, Utcubamba (25 October 1979, D. Wiedenfeld, LSUMNS 92029; Fig. 3), and three from La Lejía in NE Peru (11 & 19 March, 16 April 1925, H. Watkins, AMNH 234533, 234532, and 234535). All four specimens show obvious intergradation between these two taxa, particularly in bill coloration (see Fig. 3 and Haffer, 1974, Fig. 16.8), and all four are males (these individuals were not included in the morphometric analyses). In addition, there are another five specimens that seem to show evidence of intergradation to a lesser degree, two females that are phenotypically mostly A. “p.” cyanolaemus (both H. Watkins: La Lejía, 19 March 1925, AMNH 234534; and Uscho, Dept. Amazonas, 3 October 1925, AMNH 234531), and three males that are phenotypically A. “p.” atrogularis but seem to have some A. “p.” cyanolaemus influence (e.g., primaries edged in russet). These latter three birds are from three localities: Divisoria, Cordillera Azul, Dept. Huanuco (17 August 1967, J. P. O’Neill, LSUMNS 62227), Huanhuachayo, Dept. Ayacucho (6 May 1971, J. P. O’Neill, LSUMNS 69410), and Abra Divisiona, Dept. Loreto (14 Aug 1977, J. W. Eley, LSUMNS 84550). It is of interest that this evidence of hybridization occurs between the subspecific pair with the fewest morphometric differences (Table 4) and close genetic affinity (Puebla-Olivares et al., 2008).

Morphometrics and hybridization in a genetic context

While divergent selection should produce phenotypic differences between species, this observation makes predictions neither in direction nor degree as far as morphometrics are concerned (i.e., it only predicts accumulating differences). Nevertheless, where differences occur between groups on a phylogeny, and the genetic distances involved, might be of further aid in inferring species limits—qualified, of course, by the many known ways in which mtDNA can be misleading about species limits (Avise & Wollenberg, 1997; Irwin, 2002; Funk & Omland, 2003; Degnan & Rosenberg, 2006; Cheviron & Brumfield, 2009; Galtier et al., 2009; Ribeiro, Lloyd & Bowie, 2011; Toews & Brelsford, 2012; Pavlova et al., 2013; Peters et al., 2014; Dolman & Joseph, 2015; Morales et al., 2015). The mtDNA topology of the phylogenetic relationships among the six major subspecific groups is given in Fig. 4 (after Puebla-Olivares et al., 2008). To examine the results of my study in relation to what is presently known about relationships among and genetic distances between the groups examined, I downloaded the mtDNA data of Puebla-Olivares et al. (2008) from GenBank, concatenated and aligned the sequences using Geneious (ver. 7.1; Kearse et al., 2012), and calculated genetic distances between the groups for which I made pairwise comparisons using MEGA (ver. 6; Tamura, Nei & Kumar, 2004; Tamura et al., 2013). Three of these groups are not monophyletic in their mtDNA (not uncommon; see Funk & Omland, 2003), but I treated the haplotypes of A. “p.” atrogularis that have introgressed into A. “p.” albivitta (see discussion below) as A. “p.” atrogularis for the calculation of genetic distances.

Figure 4 The mtDNA topology of the relationships among the six major subspecific groups, following Puebla-Olivares et al. (2008).

Taxa labeled with a “(+)” are non-monophyletic in mtDNA. Values between the major subspecific groups are the between-group mean genetic distances between them.

Phenotypic evidence of hybridization occurs only between the most closely related pair on this tree, but the presence of A. “p.” atrogularis mtDNA in birds that are phenotypically A. “p.” albivitta with no outward evidence of hybrid characteristics indicate that gene flow can occur between groups that are on average 4.2% divergent (Fig. 4, bottom clade). Contrasting the number of morphometric differences that have accumulated between the pairwise comparisons of major groups that might hybridize due to proximity (Table 4, excluding albivitta-atrogularis) with genetic distance reveals a positive correlation (linear regression, F = 6.07, P = 0.04, R2 = 0.67; Fig. 5).

Figure 5 The relationship between genetic distance (Figure 3) and the accumulation of morphometric differences (Table 4) between the major subspecific groups that might hybridize due to proximity.

The positive correlation is that predicted by the processes of anagenesis and speciation.

Discussion

My results show that a complex array of morphometric similarities and dissimilarities occur between the major subspecific groups of A. “prasinus.” Moreover, these variations differ between the sexes. The morphometric data (Tables 3–5) clearly show that these taxa are not simply “cookie-cutter” renditions of a green toucanet bearing different head and bill colors.

It might be argued that morphological change among the major subspecific groups of A. “prasinus” are to be expected: that Bergmann’s rule of increased body size with latitude (and elevation; Mayr, 1963) would apply to populations of Aulacorhynchus “prasinus” and predispose this examination to finding morphometric differences. Analyses showed no association between male principal components and either latitude or longitude. Further, while female principal components showed a small but significant effect from latitude and longitude, only longitude by itself showed a significant association with PC1, leaving no evidence for Bergmann’s rule operating among these taxa as a group. The absence of any geographic effect in males suggests that some other factor, perhaps sexual selection, overcomes the relatively small geographic effect that otherwise occurs in females.

Another possibility is that differences occur not due to genetic disjunctions among locally adapted lineages, but rather to environmental variables affecting development (e.g., James, 1983; James, 1991; West-Eberhard, 2003). If we consider these allopatric forms as a series of natural experiments in differentiation, I suggest that we can consider group-specific morphological evolutionary adaptation as a more likely basis for the observed differences than the possibility of developmental plasticity (although the latter is itself subject to selection; West-Eberhard, 2003), especially because they are coupled with color changes that are not attributable to developmental plasticity. Several clear patterns emerge from the data to suggest that a simple change in environment is not the cause of the morphological differences occurring among these major groups. These patterns include sexually different morphometric changes between groups (Table 4), a large-small-large pattern going from northern Middle America to South America (Table 4), and discordant changes between PC1 and PC2 (Table 5). Nor are the differences among them simply differences in size, as Bergmann’s rule would predict; indeed there is a minimal influence of geography alone (latitude and longitude; Tables 4 and 5). If between-group differences were driven by developmental plasticity, I would expect more evidence of underlying predictable patterns, such as sexually similar responses. Instead, heterogeneity is the hallmark of the differences observed, and ecological adaptation is (hypothetically) a reasonable explanation (see also Mayr, 1963; Price, 2008).

Thus, in A. prasinus sensu lato we have complex morphometric changes occurring in conjunction with a series of additional complex phenotypic changes, for example: paedomorphic basal bill encrustations retained and enhanced in adult A. “prasinus” wagleri (an important character that alone among these examples is not simply one of color); chestnut coloration in the bill of A. “p.” albivitta; changes in coloration at the base of the bill among the groups, and a double leapfrog pattern in throat colors (light-dark-light-dark; Fig. 1). Concordant shifts in suites of mensural and other morphological characters are precisely what we would predict to occur between individuals representing genetically disjunct, locally adapted gene pools. Consequently, this evidence suggests that this is what they are, and at these levels of morphological differentiation (morphometrics, coloration, and pattern) we would usually consider these groups to be full biological species. But that conclusion does not include all of the evidence available.

Haffer (1974), who measured 66 A. prasinus (sensu lato), treated all forms as subspecies. Given evidence of hybridization in two cases in the A. “prasinus” complex, he concluded that differences would probably not prevent interbreeding; he thus retained the post-Peters (1948) taxonomy for this group. The evidence of intergradation occurs between A. “p.” atrogularis and A. “p.” cyanolaemus in Peru (a dataset that I have expanded upon above) and seemingly rather widespread intergradation among A. “p.” albivitta forms (A. “p.” albivitta, phaeolaemus, and griseigularis). Probably to simplify his survey of the entire family, Haffer (1974) generally treated all subspecies as equivalent in degree of differentiation, not making distinctions between minor and major variants. Thus, his correct observation of apparently pronounced gene flow among A. “p.” albivitta forms may have overshadowed the comparative rarity of gene flow among the major forms.

Short & Horne (2001) also pointed to intergradation among subspecies and noted (p. 326) that “differences in colour of head and of the bill seem ineffective in preventing interbreeding, e.g., in NW South America; allopatric taxa are no more distinctive in features than are the interbreeding forms (the entire complex ought to be studied carefully before any one taxon is elevated to the status of species).” It is likely that the evidence of hybridization discussed by Haffer (1974) and Short & Horne (2001) formed the basis for Peters (1948) massive lumping, although he gave no reasoning.

Navarro et al. (2001) studied the phenotype of the A. “prasinus” complex, examining 247 specimens from Middle America and 58 from South America. Unlike my study, they included in their analyses patterns and colors of the head and bill. They sidestepped the issue of hybridization and concluded that there were four species in Middle America (A. wagleri, prasinus, caeruleogularis, and cognatus) and three more in South America (A. lautus, albivitta, and atrogularis (“nigrogularis” in their abstract is an error)).

Puebla-Olivares et al. (2008) provided the first genetic data for the A. “prasinus” complex. My conclusions, which I will detail below, are mostly congruent with theirs, but there are also key differences (Table 1). These differences stem mainly from how we choose to interpret the genetic data and morphological diagnosability. For example, Puebla-Olivares et al. (2008) relied heavily on genetic distance, reciprocal monophyly, and inferred gene flow using relatively small population samples and a single locus (mtDNA). Although the evolutionary hypothesis that their data provides for this group is likely to be mostly accurate, the power of these data for determining species limits is not high, particularly in a group in which hybridization has played a pivotal role in determining taxonomy. Moreover, if we set aside genetic distance for a moment, there are other named, allopatric forms that are morphologically diagnosable and reciprocally monophyletic in their data that they did not highlight as being likely species (e.g., the subspecies A. prasinus warneri, volcanius, and chiapensis within prasinus, sensu stricto). Further, Puebla-Olivares et al. (2008) showed two A. “p.” albivitta from NE Ecuador in their A. “p.” “atrogularis” clade with no discussion (contrast their Table 1 locality 22 with their Figs. 1 and 2; A. “p.” albivitta is the form that occurs in NE Ecuador, not A. “p.” atrogularis, given incorrectly in their Fig. 1 but correctly in their Table 1). This is a clear mismatch of morphology and genetics: two individuals that are phenotypically A. “p.” albivitta (catalogued as such and verified by me from photographs) have mtDNA haplotypes more closely related to A. “p.” atrogularis and cyanolaemus. (The vouchers are ANSP 185,312 for tissue 4837 and ANSP 185,311 for tissue 4799; only tissue numbers are given for these birds by Puebla-Olivares et al. (2008).) This produces a paraphyletic A. “p.” albivitta and suggests that there has been historical gene flow between forms that are quite different.

While these data are important, using them to determine species limits is problematic. And considering morphology, their observation (Puebla-Olivares et al., 2008:47) that the diagnostic morphological attributes of their focal clades “could facilitate reproductive isolation” is unduly optimistic given evidence to the contrary (e.g., their own unremarked A. “p.” albivitta results and the quote of Short & Horne (2001) above). Genetic distance is not a reliable indicator of species limits in birds (Price, 2008; Winker, 2009). A better indicator is how the birds themselves interact when in contact (Mayr, 1969; Mayr & Ashlock, 1991). And when examining gene flow, sample sizes and geographic coverage become critically important (Winker, 2010), especially in groups, such as the A. “prasinus” complex, whose taxonomy has been so affected by evidence of hybridization.

Hybridization

Fortunately, with respect to gene flow we do have larger sample sizes if we use diagnostic morphological attributes as a surrogate (i.e., the plumage and bill characteristics upon which the named forms have been based). Despite what is likely to be ample opportunity for gene flow through dispersal across the five zones of contact or “nearest approach”, evidence of hybridization among the major groups in the A. “prasinus” complex presently exists in just two cases: between A. “prasinus” atrogularis and cyanolaemus (phenotypic only, as detailed above), and in the genetic results of Puebla-Olivares et al. (2008), which showed two A. “p.” albivitta specimens from NE Ecuador with mtDNA more closely related to A. “p.” cyanolaemus/atrogularis. This latter case suggests that historical crossing may have occurred across this zone, although morphological evidence of this is not yet evident. Aulacorhynchus “prasinus” is uncommon in this region, and the ranges of A. “p.” albivitta and cyanolaemus are not known to come into contact (Ridgely & Greenfield, 2001; Restall, Rodner & Lentino, 2006). These two A. “p.” albivitta specimens also demonstrate that phenotypic evidence of hybridization, which does occur and has been useful in past evaluations in A. “prasinus”, can be absent despite gene flow (but even nuclear genomic evidence of hybridization can disappear over a few generations of backcrossing; Lavretsky et al., 2016).

From this dataset, therefore, we know that hybridization in toucanets can be visible and invisible, the latter probably after repeated backcrossings to one parent taxon. We might, however, consider the visible hybrids to be roughly indicative of a hybridization rate. Current evidence thus suggests that hybridization between A. “p.” albivitta and cyanolaemus is rare. In the case of A. “p.” cyanolaemus and atrogularis, however, my results expand the scope of hybridization recognized, both in number of possible F1 specimens (those most intermediate in characters) and in the broader distribution of specimens likely exhibiting intergradation (and note that it is bi-directional). The fact that all four putative F1 specimens are males suggests the possibility that the two forms are sufficiently divergent that genetic incompatibilities are preventing viability of the heterogametic sex, which is the female in birds (Haldane’s rule; Price, 2008). However, this is not a significant departure from the sex ratio of the rest of the A. “prasinus” sample I examined (P = 0.13, Fisher’s exact test), and the two forms are not very divergent genetically in mtDNA (<1%; Fig. 4 and Puebla-Olivares et al., 2008). The matrilineal passage of A. “p.” cyanolaemus/atrogularis mtDNA into A. “p.” albivitta in NE Ecuador (Puebla-Olivares et al., 2008) also suggests that Haldane’s rule is not operating among major subspecific groups in South America. In the case of A. “p.” cyanolaemus and A. “p.” atrogularis, hybrids and intergrades represented a substantial percentage of the specimens I was able to examine of these taxa (as many as 10 specimens, with remaining sample sizes of 19 each of cyanolaemus and atrogularis). Given the small sample, it remains unclear whether this reflects the true incidence of hybrids between these taxa, but given present evidence it is substantial.

Species limits

Despite considerable combined evidence from coloration, morphometrics, and mtDNA data, comprehensive and accurate species limits for this group remain elusive, no matter what species concept one chooses to use. Using the biological species concept, I suggest that consideration of all of the available evidence indicates that we should recognize five species in the A. “prasinus” complex (A. wagleri, prasinus, caeruleogularis, albivitta, and atrogularis), each with any associated named subspecies (Appendix). Further study could raise this number (e.g., by splitting A. atrogularis cyanolaemus from atrogularis again). Under a phylogenetic species concept (PSC), one could probably raise every allopatric population in A. “prasinus” to the species level, resulting in at least 12 taxa using morphology alone. So how many species of toucanets are there in the A. “prasinus” complex?

Historically, evidence of hybridization has often driven taxonomic decisions under the biological species concept (BSC), as has apparently occurred in this case. My interpretation of the taxonomic history of Aulacorhynchus “prasinus” is that evidence of hybridization and intergradation between named forms (among forms of the A. “p.” albivitta group and between A. “p.” albivitta and cyanolaemus) caused all named forms to be lumped together as one species (Peters, 1948). But hybridization is not uncommon between full species (Grant & Grant, 1992), and avian taxonomists have used a working definition of the BSC that recognizes this (e.g., Short, 1969; Johnson, Remsen Jr & Cicero, 1999; Winker et al., 2007). Gene flow, reproductive isolating mechanisms, and lineage reticulation remain fundamentally important evolutionary phenomena affecting species diversity and the process of evolutionary divergence, and thus they require consideration. Effective lineage reticulation requires that hybrid offspring have equal or greater fitness than offspring of pure parental forms. Also, gene flow must occur frequently enough to overcome the differentiating selective factors likely to be operating on largely allopatric populations (and this relationship is nonlinear; see Winker, 2010 for discussion). The more differences there are between populations in morphology, the more differences there are likely to be in selective factors operating on these populations and the more difficult effective gene flow is likely to be between populations; at larger scales this results in the general correlation between morphological difference and reproductive isolation (Mayr, 1963; Price, 2008).

Classic systematics and taxonomy (Mayr, 1969; Mayr & Ashlock, 1991) uses a comparative approach to determine species limits among allopatric taxa, examining what occurs at contact zones (if available) and/or what occurs in similar cases in closely related taxa. In previous work on A. “prasinus” taxonomy I do not think enough credit has been given to the dispersal abilities of these birds. And yet despite that ability there is a lack of evidence for gene flow (using phenotype as an indicator) between five of the major subspecific groups (A. prasinus-wagleri, prasinus-caeruleogularis, caeruleogularis-albivitta, albivitta-cyanolaemus). For example, in south-central Mexico (Oaxaca), A. prasinus and A. wagleri breed within about 100 km of each other, a distance that A. prasinus individuals appear to move routinely away from their breeding areas, e.g., at the base of the Yucatan Peninsula (e.g., Land, 1970; Jones, 2003), which does not seem unusual for an arboreal frugivore (see also discussions in O’Neill & Gardner, 1974, and Navarro et al., 2001). Hybridization per se is not sufficient evidence for conspecificity, and in this group I find the lack of hybrids at most zones of potential crossing of major subspecific groups to be more compelling in the determination of species limits than its clear and seemingly routine presence at one—particularly in light of the repeated evidence of varying suites of morphological characters changing abruptly across these zones. However, I do consider that the apparent frequency of hybridization between A. atrogularis cyanolaemus and A.a. atrogularis warrants a conservative approach to their separation at the species level, and thus I do not recommend doing so without more evidence. In short, morphologically there is no evidence for hybridization between five of the major subspecific groups, despite likely opportunity, especially in northern Middle America. This is coupled with pronounced morphometric differences between these groups, suggesting group-specific ecological adaptation in addition to whatever social selection factors have likely caused the rather dramatic head and bill color differences (Fig. 1 and Table 2).

The populational processes of lineage divergence and the hierarchical nature of differentiation that accrues as gene flow decreases and divergent selection produces increasingly different phenotypes (anagenesis) have produced gradations of differentiation in the genus Aulacorhynchus. This is seen from population-level differences of little significance (e.g., among some questionably recognizable subspecies; see Appendix), to diagnosable isolated populations within biological species, to full biological species, to a recognizable subgeneric group (members of A. “prasinus” sensu lato). Genetic data also support this subgeneric group (Puebla-Olivares et al., 2008), and the name Ramphoxanthus, which is particularly fitting (i.e., yellow-bills), is available for it (Bonaparte, 1854). This group diverged from other members of the genus during the Miocene (approximately 6–9 million years ago; (Bonaccorso & Guayasamin, 2013). Within the five groups comprising A. “prasinus” that I consider full biological species (A. wagleri, prasinus, caeruleogularis, albivitta, and atrogularis) there are a number of diagnosable subspecific taxa that are clearly evolutionarily significant units, and some, being 100% diagnosable, could be called phylogenetic species (e.g., A. p. warneri, A. c. cognatus, A. albivitta lautus). These, however, do not represent major phenotypic differences, and I consider this continued lumping to be warranted given present evidence, which includes hybridization and intergradation between other subspecies with similar degrees of differentiation within the major subspecific A. “prasinus” groups (e.g., A. p. prasinus-virescens, A. albivitta phaeolaemus-griseigularis; Short & Horne, 2001 and pers. obs.).

Voice has not diverged among these five groups as much as it has in other species in the genus (Schwartz, 1972; Short & Horne, 2001). Indeed, Short & Horne (2001:327) related that “Calls of forms in Peru, Venezuela, Panama, Costa Rica and Mexico are much alike...” However, more work is warranted in this area. For example, A. wagleri has a slower pace to its vocalizations than A. prasinus (from www.xeno-canto.org, 4 A. wagleri average 1.85 calls/sec while 6 A. prasinus average 2.13 calls/sec; XC 274798, 219401-2, and 177515 vs. XC 96724, 256673, 256311, 233097, and 138132-3). Other differences may be apparent with increased sample sizes.

Considering my suggested taxonomy (Appendix) in relation to the mtDNA tree of Puebla-Olivares et al. (2008), there are two paraphyletic species (Fig. 4). First, A. albivitta is paraphyletic with respect to A. atrogularis-cyanolaemus. Second, my treatment of A. caeruleogularis is paraphyletic with respect to A. prasinus and A. wagleri because it includes A. “p.” cognatus; that has been the norm since its description, only Navarro et al. (2001) and Puebla-Olivares et al. (2008) have treated it as a full species thus far. Paraphyletic species are not uncommon (Funk & Omland, 2003; Oyler-McCance, St. John & Quinn, 2010), but there is clearly work remaining to be done on species limits in this complex, especially in South America. For example, the distributions of A. atrogularis cyanolaemus, A. a. atrogularis, and the hybrid zone between them warrant further study, as does the apparently rare instance of crossing between A. albivitta and A. atrogularis cyanolaemus in Ecuador (mtDNA evidence of (Puebla-Olivares et al., 2008). Also, the relationship between A. c. caeruleogularis and A. c. cognatus in Panama bears further investigation; they are phenotypically relatively similar (Short & Horne, 2001:325 also noted their close resemblance) in contrast to, e.g., A.a. atrogularis and cyanolaemus. Larger sample sizes, more loci, coverage of hybrid zones, and continued recognition that there are relatively major and minor phenotypic variants among these named taxa will be needed to finally and fully resolve species limits in this group.

Supplemental Information

Data S1 Aulacorhynchus prasinus complex all data

Click here for additional data file.

Data S2 Aulacorhynchus prasinus complex no missing data matrix with PCA

Click here for additional data file.

I thank the curators and staff of the following collections for their kindness during visits or for loaning material: Louisiana State University Museum of Natural Science (LSUMNS), American Museum of Natural History (AMNH), Southwestern College (SWC), National Museum of Natural History (USNM), Carnegie Museum (CM), Field Museum of Natural History (FMNH), Moore Laboratory of Zoology (MLZ), the Academy of Natural Sciences of Philadelphia (ANSP), Museo de Zoologia, Facultad de Ciencias, Universidad Nacional Autonoma de México (MZFC), and the Colección Nacional de Aves, Instituto de Biología, Universidad Nacional Autonoma de México (CNAV). John O’Neill, Matthew Miller, Alexandre Aleixo, and several anonymous reviewers gave helpful comments on previous drafts of the manuscript. Rob Faucet and Max Thompson helped with localities. Nate Rice provided photographs of the ANSP vouchers of A. albivitta with A. atrogularis/cyanolaemus haplotypes in Puebla-Olivares et al. (2008). Finally, I wish to thank Kristin Williams for her excellent painting of the A. “prasinus” group.

Appendix

Suggested taxonomy. Because I have examined all of the described taxa in the complex, this revision includes subspecies (although quantitative analyses were not undertaken below the level of the six major groups). Given below are species, subspecies, authors of original descriptions, type localities, and notes pertaining to each species. Distribution is not included, because I did not examine all existing specimens and can add little of substance to distributions set forth by the authors cited herein. The species sequence given follows the relationships in the mtDNA tree of Puebla-Olivares et al. (2008) but with the two major clades flipped to better accommodate the group’s geographic distribution (as I have also done in Fig. 4).

Genus Aulacorhynchus (green toucanets), subgenus Ramphoxanthus

Aulacorhynchus wagleri (Sturm, 1841 in Gould, 1841–47:pl. 16 (heft 2, pl. 6)). Wagler’s Toucanet.	
no type loc. [= Guerrero and Oaxaca, Mexico].	
Aulacorhynchus prasinus (Gould, 1833). Northern Emerald Toucanet.	
A. p. prasinus (Gould, 1833). Mexico [= Valle Real, Oaxaca].	
A. p. warneri Winker (2000). Volcán San Martín, Sierra de Los Tuxtlas, Veracruz, Mexico.	
A. p. virescens Ridgway (1912:88). Chasniguas, Honduras.	
A. p. volcanius Dickey & Van Rossem (1930:53). Volcán de San Miguel, Dept. San Miguel, El Salvador.	

Notes: A. p. stenorhabdus (Dickey & Van Rossem, 1930:52) and A. p. chiapensis (Brodkorb, 1940) are considered synonyms of A. p. virescens; variation among them appears to be clinal (see also Monroe Jr, 1968). Wetmore (1941, notes in USNM) considered chiapensis as “doubtfully separable”, but recognized stenorhabdus. See notes under A. albivitta regarding the English common name.

Aulacorhynchus caeruleogularis (Gould, 1854:45). Blue-throated Toucanet.	
A. c. caeruleogularis (Gould, 1854:45). Veragua [, Panama] [= Boquete, Chiriquí; (Wetmore, 1968:508].	
A. c. cognatus (Nelson, 1912:4). Mount Pirri (at 5,000 ft altitude) head of Rio Limon, eastern Panama.	

Notes: A. c. maxillaris (Griscom, 1924:2) is considered a synonym of A. c. caeruleogularis (cf. Wetmore, 1968:509). See Wetmore (1968) for citation of the name caeruleogularis appearing first in the Zoologist in 1853; no description appears there, however, the reference being a report of what occurred at two meetings in February 1853 (D.W.M., 1853). Olson (1997) provided more notes on these occurrences in relation to Gould.

Aulacorhynchus albivitta (Boissonneau, 1840:70). Southern Emerald Toucanet.	
A. a. lautus (Bangs, 1898:173). San Miguel [, Sierra Nevada de Santa Marta], Colombia.	
A. a. griseigularis Chapman (1915:639). Santa Elena (alt. 9,000 ft.), Cen. Andes, Antioquia, Col.	
A. a. phaeolaemus Gould (1874:184). Concordia, in Columbia [sic], and Merida, in Venezuela [= Concordia, Antioquia, western Colombia; Hellmayr, 1911:1213].	
A. a. albivitta (Boissonneau, 1840:70). Santa-Fé de Bogota [, Colombia].	

Notes: Chapman (1917) inexplicably omitted the occurrence of the species (endemic subsp. lautus) in the Santa Marta region. More detailed study is needed to resolve problems in the status, relationship, distributions, and nomenclature of phaeolaemus and griseigularis (see Chapman, 1917; Haffer, 1974). The English name for this species given by Cory (1919:377), White-throated Toucanet, is only appropriate for the subspecies albivitta, and thus is more appropriate at the species level for A. prasinus (sensu stricto, though not used there). The other subspecies of albivitta are all grayish or grayish-blue on the throat. Del Hoyo & Collar (2014) suggested Grayish-throated, but this overlooks both white-throated birds and those with blue in the throats. Accordingly, I have suggested more fitting English names for this species and A. prasinus.

Aulacorhynchus atrogularis (Sturm, 1841 in Gould, 1841–47:heft 2, pl.2 & text). Black-throated Toucanet.	
A. a. cyanolaemus Gould, 1866:24. Loxa [=Loja] in Ecuador.	
A. a. atrogularis (Sturm, 1841 in Gould, 1841–47:heft 2,pl.2 & text). Andes of Peru [=Chunchamayo, central Peru; (Cory, 1919:380).	
A. a. dimidiatus (Ridgway, 1886:93). No loc.; suggested by O’Neill & Gardner (1974:703) to be along the eastern foothills of the Andes of central southern Peru.	

Note: Recognition of A. a. dimidiatus follows O’Neill & Gardner (1974). A. a. cyanolaemus is blue-throated (Fig. 1).

Additional Information and Declarations

Competing Interests

Author Contributions

Animal Ethics

Data Availability

The author declares there are no competing interests.

Kevin Winker conceived and designed the experiments, performed the experiments, analyzed the data, contributed reagents/materials/analysis tools, wrote the paper, prepared figures and/or tables, reviewed drafts of the paper.

The following information was supplied relating to ethical approvals (i.e., approving body and any reference numbers):

This work involved museum specimens of vertebrate animals, none of which were collected for this study, and so no permits or permissions were required.

The following information was supplied regarding data availability:

The raw data have been supplied as Supplementary Files.

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
