# Peer review of "An examination of species limits in the Aulacorhynchus “prasinus” toucanet complex (Aves: Ramphastidae)"

_PeerJ, doi:10.7717/peerj.2381_

## Round 0.1 · original submission · Major Revisions

Both reviewers provided very thoughtful and important issues that have to be addressed before the paper is ready for publication. Pay particular attention to the relationship between the morphological variation found in your study and previous phylogenies that have been published before on the "prasinus" group.

Please also note the extensive annotated document from Reviewer 1

Reviewer 1 ·

Basic reporting

- There are some parts of the text that are not clear enough and must be rewritten. Besides, the author did not make explicit some relevant information. Please see the annotations added to the manuscript PDF.

- There is an issue concerning the Sections subdivision (Methods and Results are presented together). Please see this comment in the manuscript PDF.

- There are some issues with the labels of tables and figures that difficult the full understanding of what it is presented. Please see the annotations added to the manuscript PDF.

Experimental design

No comments

Validity of the findings

No comments

Annotated reviews are not available for download in order to protect the identity of reviewers who chose to remain anonymous.

·

Basic reporting

I found the paper rather too long and not easy to follow.
But what made it really hard on me was that species limits were evaluated primarily through morphometrics and that species groups were delimited only based on statistically significant differences between PCA values.
I believe that this paper adds some important knowledge on the "prasinus" group, but the morphometric data should be discussed in more direct perspective with the phylogenetic data available so far from the different papers cited. In other words, what about to contrast the significant PC pairwise value differences against genetic distances? I would be more confortable accepting purely phenetic results if I knew that they had some bearing on some tree based hypothesis of ancestry. This comparison is made throughout the paper, but it should be the main part of the results in my view.
Even if you do not contrast the morphometric data directly with one or several trees, the paper fails to provide lists of sets of discrete characters (plumage for instance) that diagnose unequivocally each different groups revealed by significant PCA value differences. In other words, is there any additional data (characters) corroborating the PCA significant differences? What are they? Do the purported hybrids between these groups vary accordingly to what is expected for the color characters? How were they considered intermediate?
In conclusion, I found this paper important because of the sheer amount of data it brings in to the subject, but it should be more concise and, if a new treatment for the complex is suggested, more detailed diagnoses should be provided as well as how they relate to already published phylogenies on the "prasinus" group.

Experimental design

No comments.

Validity of the findings

No comments.

---

## Round 0.2 · Minor Revisions

I'm glad to see that many of the suggestions by the reviewers have been addressed, but there are still a few issues (in the attached PDF) that need to be tackled before the paper is ready for submission.

Reviewer 1 ·

Basic reporting

There are only minor issues, concerning figure labels and some isolated suggestions along the text. Please see the annotations added to the manuscript PDF.

Experimental design

No comments

Validity of the findings

No comments

Annotated reviews are not available for download in order to protect the identity of reviewers who chose to remain anonymous.

---

## Round 0.3 · accepted · Accept

I believe that you properly addressed all of the comments by the reviewer. Congratulations!